# Experimental Investigation of Reynolds Number and Spring Stiffness Effects on Vortex-Induced Vibration Driven Wind Energy Harvesting Triboelectric Nanogenerator

**DOI:** 10.3390/nano12203595

**Published:** 2022-10-13

**Authors:** Qing Chang, Zhenqiang Fu, Shaojun Zhang, Mingyu Wang, Xinxiang Pan

**Affiliations:** 1Marine Engineering College, Dalian Maritime University, Dalian 116026, China; 2School of Navigation and Shipping, Shandong Jiaotong University, Weihai 264200, China; 3School of Electronics and Information Technology, Guangdong Ocean University, Zhanjiang 524088, China

**Keywords:** vortex-induced vibration, triboelectric nanogenerator, wind energy, spring stiffness

## Abstract

Vortex-induced vibration (VIV) is a process that wind energy converts to the mechanical energy of the bluff body. Enhancing VIV to harvest wind energy is a promising method to power wireless sensor nodes in the Internet of Things. In this work, a VIV-driven square cylinder triboelectric nanogenerator (SC-TENG) is proposed to harvest broadband wind energy. The vibration characteristic and output performance are studied experimentally to investigate the effect of the natural frequency by using five different springs in a wide range of stiffnesses (27 N/m<K<90 N/m). The square cylinder is limited to transverse oscillation and experiments were conducted in the Reynolds regime ($3.93\times 10^{3}–3.25\times 10^{4}$). The results demonstrate the strong dependency of VIV on natural frequency and lock-in observed in a broad range of spring stiffness. Moreover, the amplitude ratio and range of lock-in region increase by decreasing spring stiffness. On the other hand, the SC-TENG with higher spring stiffness can result in higher output under high wind velocities. These observations suggest employing an adjustable natural frequency system to have optimum energy harvesting in VIV-based SC-TENG in an expanded range of operations.

## 1. Introduction

With the rapid development of human society, various processing and service terminals are continuously produced and put into society to meet people’s growing demands for life and production. As the number and size of these scattered terminals grow, people began to seek a way to manage them more efficiently, and the concept of the Internet of Things came into being [1]. The information acquisition method proposed by the Internet of Things is inseparable from the support of sensors. The early Internet of Things is also called the sensor network. These sensors are so widely distributed that the power grid is often unable to meet their needs. Most sensors are powered by batteries. This traditional power supply method has obvious disadvantages. The power of the battery will be exhausted after a certain period of time. At this time, it is very labor-intensive and time-consuming to replace the battery, especially for some sensors located in remote areas. Chemical energy storage batteries and non-rechargeable disposable batteries will cause huge pollution to the environment during production and scrapping. A power supply method that can provide power for distributed sensor networks for a long time is of great significance for the development of the Internet of Things technology and has become a hot spot in the field of wireless sensor network research.

There are many kinds of energy contained in the environment, such as solar energy, thermal energy, tidal energy, wind energy, and so on. Compared with other types of environmental energy, wind energy has its unique advantages. Wind energy is present in nearly every corner of the planet. It is not affected by rain or shine, and can often present as fairly regular mechanical energy, with a high enough energy density to support the normal operation of wireless sensor nodes [2]. The electromagnetic generator (EMG) and piezoelectric generators are the most common forms to harvest wind energy [3,4,5,6,7,8,9]. However, they are not suitable for powering small, distributed electronics due to their structure, cost, and output. Designing a miniature wind energy harvesting device to harvest ambient wind energy and convert it into electricity provides a new solution to self-powered wireless sensor networks.

Triboelectric nanogenerators (TENG) can efficiently convert low-frequency mechanical energy into electricity in environments that are difficult to harvest with conventional EMGs [10]. TENG has strong scalability [11,12,13,14,15,16,17] and excellent development prospects in the field of wind energy harvesting technology [18,19,20,21]. Rotary vane type and flutter film type are the most common research directions that combine TENG with wind energy harvesting [22,23,24,25,26,27,28,29]. Li et al. [30] designed a breeze-driven wind scoop triboelectric nanogenerator (BD-TENG) to harvest wind energy to power smart agricultural IoT sensors. By choosing lightweight rotor materials and designing a suitable wind scoop structure, the critical speed of BD-TENG can be as low as 3.3 m/s. When the wind speed is 4 m/s, the output voltage of BD-TENG is 330 V, the current is 7μA, the transferred charge is 137 nC, and the peak power is 2.81 mW. The BD-TENG can operate normally even in low-wind speed environments and harvest wind energy efficiently. Zhao et al. [31] developed a flag-type triboelectric nanogenerator (F-TENG) that can harvest wind energy and studied its networking characteristics. The flag-type triboelectric nanogenerator array based on flutter can enhance its power generation performance significantly. The array of F-TENGs shows great potential for applications in wind energy harvesting and power supply for wireless sensors. Wind energy can be effectively harvested and utilized by micro-wind energy harvesting devices with different principles in various forms. However, the rotary vane type TENG has a high material wear rate. Due to the existence of the rotary mechanism, it is easy to cause mechanical failure and increase the maintenance cost. The output of a flutter film triboelectric nanogenerator is easily affected by the incoming flow and the surrounding environment and becomes unstable [32]. There is a strong demand to design a triboelectric nanogenerator for wind energy harvesting with a simple structure and stable output.

Vortex-induced vibration (VIV) is a common fluid-structure interaction phenomenon in the field of fluid mechanics. The vibration is caused by the interaction of the vortex shedding and the bluff body. When the fluid flows through the bluff body, the vibration of the bluff body is called vortex-induced vibration [33,34]. This phenomenon can convert wind energy into considerable mechanical energy. Vortex-induced vibration generators based on the principles of piezoelectric and electromagnetic are the most common forms [35,36,37]. These works provide a great reference value for the development of vortex-induced vibration triboelectric nanogenerators for wind energy harvesting. Recently, Wang et al. [38] proposed a wind energy harvesting TENG, which is based on vortex-induced vibration. VIV-TENG exhibits the characteristics of being waterproof and having high output performance. They systematically studied the effect of the system mass ratio on the amplitude ratio and the lock-in region. Based on the specific stiffness spring that they used, a general conclusion could not be made. Although the natural frequency of the system is an essential parameter in analyzing the vortex-induced vibration, the investigations on the effect of spring stiffness on the behavior of a square cylinder in VIV have not been well documented.

Herein, a vortex-induced vibration-driven square cylinder triboelectric nanogenerator (SC-TENG) is proposed and investigated. The occurrence and development process of the square cylinder vibration under different wind velocities was explored through the smoke wire visualization experiments. The effects of spring stiffness and flow velocity on the amplitude response of a square cylinder in one degree of freedom are experimentally studied. Five different linear springs were used to better explore the effect of the natural frequency of the system on the vibration and output behavior of the SC-TENG. The results from this work can be used by other researchers to design adjustable stiffness systems based on the flow velocity for extracting maximum energy from the wind.

## 2. Materials and Methods

As shown in Figure 1a, the SC-TENG consists of four parts, i.e., a square cylinder, an internal honeycomb structure power generation unit, four springs, and an external frame structure. The internal structure consists of a vibrator with honeycomb holes, upper and lower electrodes covered with copper foil, and several PTFE balls. The square cylinder and internal vibrator are printed by a 3D printer with PLA material and UV-curable resin, respectively. The size of the square cylinder is 150 mm×30 mm×30 mm. The size of the internal power generation is 120 mm×26 mm×8 mm. The diameter of the PTFE ball is 5 mm. A honeycomb structure is designed as the vibrator. As shown in Figure A1, compared with the conventional square grid, the honeycomb grid can accommodate more PTFE balls for the same area of the power generation unit. The application of the honeycomb structure improves the output performance of the SC-TENG. On one hand, the compact nature of the honeycomb structure increases the effective contact area between the PTFE balls and electrode layers. Figure A1 compares the number of grooves in the square-grid frame and the honeycomb frame. It can be seen that for the internal power generation unit with a total area of $120\times 26\;{\mathrm{mm}}^{2}$ and PTFE balls with a diameter of 5.0 mm, the honeycomb frame has 43 grooves, 15 more than the square-grid frame. As a result, the effective contact area is increased by 53.4%. On the other hand, the electrical output of the TENG is positively correlated with the effective contact area according to previous work [38]. Considering that the effective contact area is determined by the number of PTFE balls. Thus, the honeycomb structure is used here. More PTFE balls can result in more charge transfer. The honeycomb structure can accommodate 43 PTFE balls here. Vibration characteristic and electrical output performance tests are conducted in a wind tunnel as shown in Figure 1b. The open loop wind tunnel used in the experiment has a testing section, which is 1.0 m long, 0.25 m wide, and 0.25 m high. The rotating speed of the blower is controlled by an inverter, which then varies the wind velocity. Five different stiffness springs in a wide range of stiffness are adopted here to investigate the effect of the natural frequency on the vortex-induced vibration characteristic. Vibration amplitude and vibration frequency are systematically analyzed to illustrate the vibration characteristic. A linear laser displacement sensor (KEYENCE IL065, KEYENCE, Ōsaka, Japan) is used to measure the vibration amplitude. The visualization of the fluid interaction with the SC-TENG is realized by the smoke flow method. The vortex shedding is captured by the high-speed camera (FATCAM Mini UX50, PHOTRON, San Diego, CA, USA) to present its formation and development. Furthermore, the vibration frequency is calculated by subtracting the timing between taken images after a complete vibration cycle. When the fluid flows over the square cylinder, periodic shedding vortices are formed in the wake at the rear of the square cylinder. If the frequency of the vortex shedding $f_{v}$ matches the natural frequency $f_{n}$ of the system, the square cylinder will vibrate up and down as depicted in Figure 1c. The internal power generation unit vibrates with the square cylinder. Meanwhile, the PTFE balls bounce in the honeycomb structure and contact the bottom and upper electrodes periodically. The PTFE balls are an electronegative material, and the copper is an electropositive material. When the PTFE ball make contact with the bottom copper electrode, electrons on the copper electrode are transferred to the surface of the PTFE ball based on the triboelectrification principle. PTFE ball vibrates with the square cylinder. The positive charges are transferred from the bottom electrode to the upper electrode during it bounces upward. A current is generated in the external circuit, PTFE balls completely contact with the upper electrode, and positive charges are completely transferred to the upper electrode. When the PTFE ball vibrates down with the square cylinder, positive charges are transferred from the upper electrode to the bottom electrode. A current is generated in the external circuit again. When the PTFE ball makes contact with the upper electrode, the charge transferred cycle is completed, as shown in Figure 1d. The COMSOL Multiphysics software (Version No. 5.5a, COMSOL Inc. Stockholm, Sweden) is applied to show the changing process of the potential difference as shown in Figure 1e. A Keithley 6514 (Tektronix, Beaverton, OR, USA) system electrometer is used to measure the electrical output performance of the SC-TENG.

## 3. Results

### 3.1. Smoke Wire Visualization Experiment of the SC-TENG

Vortex-induced vibration of SC-TENG can occur under wind excitation. The occurrence and development process of its vibration and its vibration characteristics determines its electrical output performance. The visualization of fluid interaction with the VIV-TENG is realized by the smoke flow method as shown in Figure 2a. Smoke wire experiment can intuitively show the motion law of the bluff body when vortex-induced vibration occurs. The main properties of the experimental setup are listed in Table 1.

The lock-in region in the VIV phenomenon can be considered like linear resonance, as the vibration amplitude increases significantly when the vortex shedding frequency $f_{v}$ becomes close to the natural frequency $f_{n}$ of the structure. In this situation, the nondimensional frequency $f^{*}=f_{v}/f_{n}$ remains close to unity [39]. The natural frequency of the SC-TENG can be expressed by Equation (1)
(1)$$f_{n}=2\pi \sqrt{\frac{K}{m_{osc}}}$$
where K is the spring stiffness and $m_{osc}=54\;g$ represents the mass of the square cylinder. Square cylinder vibration videos captured by a high-speed camera are utilized to measure the vibration frequency $f_{v}$. Reynolds number $R_{e}$ is used to present the incoming flow velocity.

As shown in Figure 2b, when the wind velocity reaches 1.6 m/s ($R_{e}=4.19\times 10^{3}$), the square cylinder starts to vibrate and maintains a small amplitude. There are obvious periodic vortices with opposite directions at the rear of the square cylinder. As the wind velocity continues to increase to 7.5 m/s ($R_{e}=1.97\times 10^{4}$), as shown in Figure 2c, the square cylinder presents the maximum vibration amplitude until the wind velocity reaches 9.5 m/s ($R_{e}=2.49\times 10^{4}$). The wind velocity continues to increase, and the square cylinder shows an unstable state with obvious torsional motion. When wind velocity exceeds 12.4 m/s ($R_{e}=3.25\times 10^{4}$), the square cylinder almost stops vibrating. The experimental phenomenon shows that when the wind velocity is in the range of 1.6–12.4 m/s ($R_{e}=4.19\times 10^{3}–3.25\times 10^{4}$), the square cylinder presents a resonance state. For the SC-TENG system with a mass ratio $m^{*}$ of 308.57, the lock-in region is 1.6–12.4 m/s. The mass ratio $m^{*}=m_{osc}/m_{d}$, here, $m_{d}$ is the displaced air mass. Within the lock-in region, there is a maximum amplitude range (7.5–9.5 m/s, $R_{e}=1.97\times 10^{4}–2.49\times 10^{4}$). Therefore, it can be intuitively seen from the smoke wire visualization experiment that with the increase of wind velocity in the lock-in region, the amplitude of SC-TENG increases continuously until it enters the maximum amplitude region. Meanwhile, it can be seen from Figure 2(bii,cii) that when the wind velocity increases, the strength of the vortex shedding forming at the rear of the square cylinder also increases. The vortex shedding interacts on the square cylinder, making the vibration amplitude of the square cylinder continue to increase when the wind velocity range is 1.6–7.5 m/s ($R_{e}=4.19\times 10^{3}–1.97\times 10^{4}$), keep the maximum vibration amplitude at 7.5–9.5 m/s ($R_{e}=1.97\times 10^{4}–2.49\times 10^{4}$), and start to attenuate when the wind velocity is 9.5–12.4 m/s ($R_{e}=2.49\times 10^{4}–3.25\times 10^{4}$) for this system.

### 3.2. Vibration Characteristics of the SC-TENG

According to the results of Modir and Goudarzi [40], spring stiffness is the key parameter that determines the vibration amplitude of the VIV system. In this section, experimental results for five different spring stiffness are presented and discussed to realize the impact of natural frequency on a one-degree-of-freedom square cylinder in VIV. Values of the spring stiffness (K) used here are listed in Table 2.

The vibration amplitude and range of the lock-in region directly determine the energy harvested by the SC-TENG from the wind. Therefore, it is necessary to explore the variation trend of the vibration amplitude and the lock-in region of the SC-TENG system when different springs are used. Figure A2 indicates the test system for conducting the vibration amplitude experiment of the system. A linear laser displacement sensor and hot wire anemometer are used to measure the displacement of the square cylinder and the wind velocity. Non-dimensional reduced velocity $U^{*}$ is used to present the incoming flow velocity versus the natural frequency and dimension of the device, which is:(2)$$U^{*}=\frac{U}{f_{n}D}$$

Amplitude ratios (A/D) versus reduced velocity and Reynolds number for different values of K are compiled in Figure 3a,b, respectively. As shown in Figure 3a, the amplitude and range of the lock-in region increase with an increase in spring stiffness. The shift to a higher operational Reynolds number increases the amplitude of vibration and the range of the lock-in region which shows the strong dependence of VIV on natural frequency. As depicted in Figure 3a, the onset of the lock-in region is more gradual for systems with higher K values. However, due to the increase of the critical velocity, the lock-in region becomes smaller for the system with higher K values. Moreover, the higher K values result in a lower maximum vibration amplitude, as shown in Figure 3a. For example, for the case with the lowest natural frequency (K=27 N/m), the cylinder vibrates two times higher than the case (K=90 N/m) at $R_{e}=2.7\times 10^{4}$. This is a key factor when a high amplitude ratio in a broad range of flow velocities is desired. As shown in Figure 3b, when using different springs, the corresponding critical reduced velocities are almost the same. Because the Strouhal number ($S_{t}$) can be considered constant for the square cylinder in the lock-in region [41]. Here, when $S_{t}=\frac{f_{v}D}{U}=\frac{1}{U^{*}}\approx 0.15$, the square cylinder starts to vibrate. As mentioned above, the VIV phenomenon occurs when the frequency ratio $f^{*}=\frac{f_{v}}{f_{n}}\approx 1$. In this work, when the square cylinder is in the lock-in region, $f^{*}\approx 0.96$ for five different springs stiffness systems. As depicted in Figure 3c, a smaller spring stiffness corresponds to a larger lock-in region.

### 3.3. Output Performance of the SC-TENG

According to the vibration characteristics of the SC-TENG, the spring stiffness is one of the factors that determine the lock-in region and critical wind velocity. The experimental apparatus is shown in Figure A3. Electrical output signals of the SC-TENG with K=27 N/m are measured under different wind velocities and shown in Figure 4a–c. The outputs increase with wind velocity until the wind velocity enters the maximum vibration amplitude range (7.5–9.5 m/s, $R_{e}=1.97\times 10^{4}–2.49\times 10^{4}$). In the maximum amplitude range, the SC-TENG can deliver the stable and maximum outputs of voltage, current and transferred charge, which can reach 110.14 V, 3.57 μA, and 38.64 nC, respectively. For the SC-TENG with K=55 N/m and K=90 N/m, the outputs show a similar trend. Furthermore, higher spring stiffness results in higher critical wind velocity, smaller working wind velocity range (lock-in region), and smaller maximum vibration amplitude range as shown in Figure 4d–i. It can be found that the spring stiffness of the SC-TENG system is different, and the output voltage and charge of SC-TENG are roughly the same. The reason is that SC-TENG is essentially a freestanding TENG. Its output voltage can be expressed by [42]
(3)$$V=-\frac{1}{C}Q+V_{OC}=-\frac{d_{0}+G}{\varepsilon_{0}S}Q+\frac{2\sigma x}{\varepsilon_{0}}$$
Here, Q and $V_{OC}$ represent the total transferred charge and open circuit voltage, respectively. C, $d_{0}$, G, and S are the total capacitance, the dielectric material effective thickness, the air gap thickness between two copper electrodes, and the contacting area size, respectively; σ denotes the charge density, x denotes the distance between PTFE balls and electrodes, and $\varepsilon_{0}$ denotes the dielectric constant. Since the triboelectric materials and device structure used are all the same, the amount of charge transferred Q is almost the same with the output voltage V. As can be seen in Table 2, higher spring stiffness corresponds to a higher natural frequency. When vortex-induced vibration occurs, the system with higher spring stiffness has a higher vibration frequency. According to $I=\frac{dQ}{dT}$, higher vibration frequency results in higher output current. The maximum output current of the SC-TENG with K=90 N/m can reach 6.12 μA. Therefore, when using SC-TENG to harvest wind energy, adjusting the spring stiffness according to the environmental conditions can make it work more efficiently. Electrical output performance is the key factor of the TENG. Table 3 lists the power density of the present SC-TENG versus those of other types of TENG-based wind energy harvesters. Apparently, the present SC-TENG is not inferior compared to those wind energy harvesting TENGs.

The size of the electronegative PTFE ball is a key factor that can affect the output performance of the SC-TENG. To investigate the relationship between the size of the ball and the output in a vortex-induced vibration system, the premise is to ensure that the vibration characteristics of the system are the same. Therefore, we used balls with diameters of 2, 3, and 4 mm for the test, while ensuring a mass ratio of 308.57 and 43 balls in the power generation unit. As is shown in Figure A4, as the ball size increases, the output voltage, current, and transferred charge increase. All PTFE balls in the power generation unit can be considered a whole. A larger size corresponds to a larger contact area with the electrode and a more transferred charge. However, it is not possible to increase the size of the ball any further while maintaining the same mass ratio and number of balls in this work. As the size of the ball increases, 43 PTFE balls cannot be accommodated by the power generation unit.

### 3.4. Demonstration Application of the SC-TENG

According to the output performance presented above, the SC-TENG can be used to harvest wind energy efficiently. Figure 5 demonstrates the output performance of the SC-TENG as a power source. As depicted in Figure 5a, the SC-TENG output power can reach 3.38 mW, corresponding to a power density of$\;135.42\;W/m^{3}$ when adopting a spring with a spring stiffness of 90 N/m at the wind velocity of 9.5 m/s. It indicates that the output power of the SC-TENG is qualified to power a small electrical appliance, such as a sensor node. Furthermore, it can be seen in Figure 5b that the 100 μF capacitors can be charged rapidly at a lower wind velocity. It exhibits good charging ability as well as working ability under low wind velocity. Figure 5c shows that the output voltage is steady over 5050 s at the wind velocity of 9.5 m/s. As can be seen from Figure 5c–e, more than 200 LEDs and a thermometer are powered by SC-TENG. The demonstration experiments show that the SC-TENG can harvest ambient wind energy to power micro electronic devices in the IoT.

## 4. Conclusions

Vortex-induced vibration of the SC-TENG is investigated experimentally in a wind tunnel at a Reynolds number of $3.93\times 10^{3}–3.25\times 10^{4}$. In this work, the focus is on identifying the effect of the natural frequency on the behavior and output performance of the SC-TENG undergoing VIV, by employing five different spring stiffness. The results including amplitude response, the range of the lock-in region of the square cylinder, and electrical output signal response in VIV are analyzed as a function of Reynolds number and reduced velocity. In addition, the process of the VIV is analyzed by smoke wire visualization technology under different wind velocities. The experiments demonstrate that SC-TENG with higher spring stiffness results in a smaller lock-in region, maximum vibration amplitude range, lower amplitude ratio, and higher critical wind velocity. The SC-TENG with K=27 N/m can obtain a maximum amplitude ration of 1.67, which is only 0.83 for K=90 N/m. According to the amplitude ratio versus Reynolds number graphs and output performance versus wind velocity graphs, it is suggested to use an appropriate spring stiffness system when it is desired to achieve maximum amplitudes and output in different flow velocities. Adjusting the system for having a lower natural frequency in lower wind velocities can help SC-TENG to have higher efficiency in a wider range of wind velocities. On the other hand, adjusting the system for having higher natural frequency can help SC-TENG to have a higher efficiency in higher wind velocity conditions.

## Figures and Tables

**Figure 1 nanomaterials-12-03595-f001:**
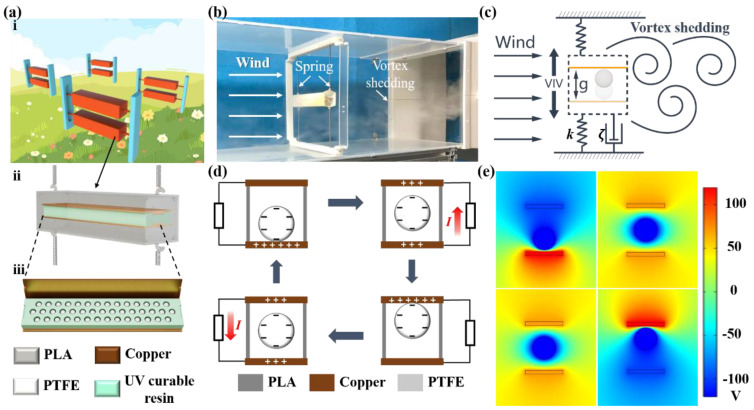
Design and working mechanism of the SC−TENG. (**a**) Schematic diagram and structure of the SC−TENG: (**i**) application scenario of the SC−TENG, (**ii**) structure of the SC−TENG, (**iii**) detailed structure of the power generation unit; (**b**) experimental setup in the wind tunnel; (**c**) vortex-induced vibration of the SC−TENG; (**d**) four working states in the process of power generation; (**e**) potential simulation diagrams of four working states.

**Figure 2 nanomaterials-12-03595-f002:**
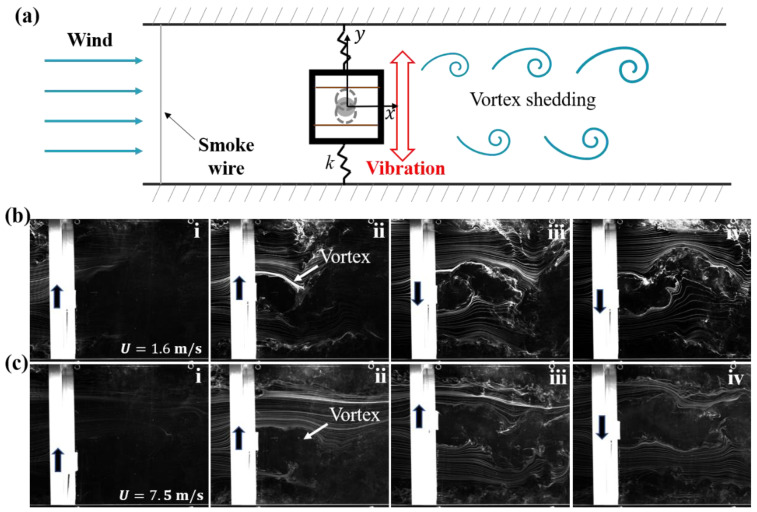
(**a**) Schematic of smoke wire visualization experimental arrangement. Smoke wire visualization of fluid interaction with the square cylinder at (**b**) 1.6 m/s, in which (**i**–**iv**) vortex development process and (**c**) 7.5 m/s, in which (**i**–**iv**) vortex development process.

**Figure 3 nanomaterials-12-03595-f003:**
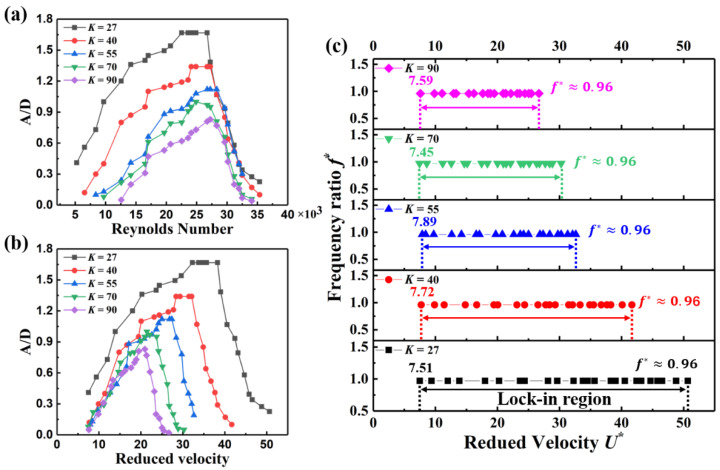
Vortex-induced vibration characteristics. (**a**) Amplitude ratio as a function of reduced velocity for different spring stiffness cases; (**b**) Amplitude ratio versus Reynolds number for different spring stiffness cases; (**c**) nondimensional frequency of response (*f*^*^) versus nondimensional velocity (*U*^*^) for different spring stiffness cases.

**Figure 4 nanomaterials-12-03595-f004:**
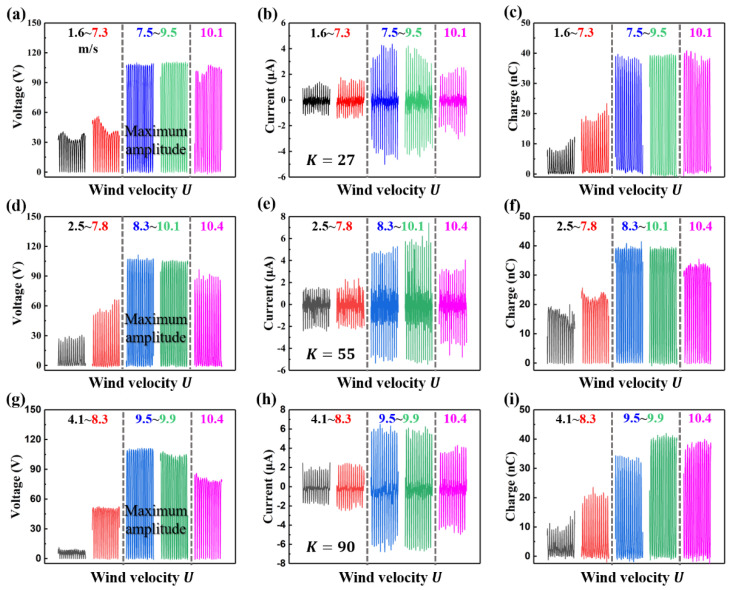
(**a**–**c**) Dependence of the output performance of the SC−TENG with K=27 N/m on the wind velocity; (**d**–**f**) dependence of the output performance of the SC−TENG with K=55 N/m on the wind velocity; (**g**–**i**) dependence of the output performance of the SC−TENG with K=90 N/m on the wind velocity.

**Figure 5 nanomaterials-12-03595-f005:**
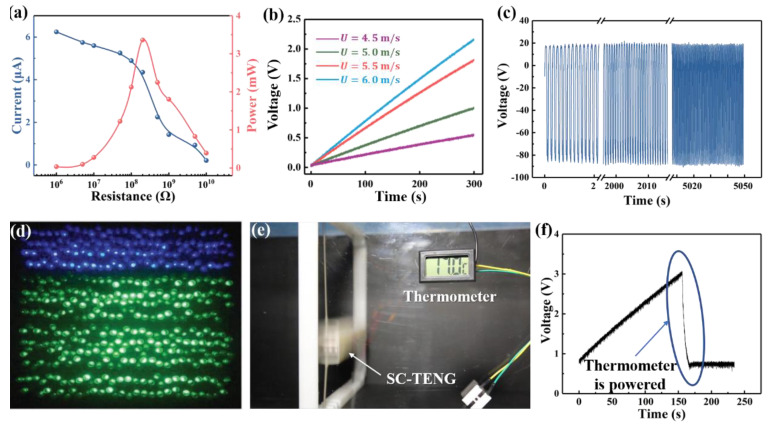
Demonstration of the SC−TENG. (**a**) The output current and the power of the SC−TENG with respect to resistance in the circuit; (**b**) charge curve of the SC−TENG under different velocities; (**c**) durability test of the SC−TENG; (**d**) more than 200 LED light bulbs are lit by the SC−TENG; (**e**) a thermometer powered by SC−TENG; (**f**) the charge-discharge curve of the SC−TENG system powering thermometer.

**Table 1 nanomaterials-12-03595-t001:** SC-TENG parameters in smoke wire visualization experiment.

Parameter	Symbol	Value
Length (mm)	L	150
Width (mm)	W	30
Height (mm)	D	30
Stiffness (N/m)	K	27
Mass ratio	$m^{*}$	308.57
Reynolds number	$R_{e}$	$3.93\times 10^{3}–3.25\times 10^{4}$
Damping factor	ζ	0.002733

**Table 2 nanomaterials-12-03595-t002:** Spring stiffness and natural frequency of five different cases used here.

Case	1	2	3	4	5
Spring stiffness (N/m)	27	40	55	70	90
Natural frequency (Hz)	8.9	10.8	12.7	14.3	16.2

**Table 3 nanomaterials-12-03595-t003:** A summary of the output performance of various wind energy harvesting TENGs.

Working Mechanism	Device	Wind Speed (m/s)	Power Density (W/m^3^)	Ref.
Vortex-induced vibration	VIV-TENG	12.4	62.2	[38]
	SC-TENG	12.4	135.42	This work
Flutter driven structure	FTEG	22	7.64	[32]
	TENG	12	88.1	[43]
	TENG	14	1.16	[44]
	TENG	Not given	39.6	[45]
	TENG	Not given	58	[46]
Rotational structure	SR-TENG	32.6	0.4	[47]
	a-TENG	8	0.19	[48]

## Data Availability

The data presented in this study are available on request from the corresponding author.

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
