# Peer review of "Experimental Investigation of Reynolds Number and Spring Stiffness Effects on Vortex-Induced Vibration Driven Wind Energy Harvesting Triboelectric Nanogenerator"

_nanomaterials, 2022, doi:10.3390/nano12203595_

Round 1
Reviewer 1 Report
The authors report a vortex-induced vibration driven square cylinder triboelectric nanogenerator (SC-TENG). The development process of the square cylinder vibration under different wind velocities is an interesting subject in this field. The effects of spring stiffness and flow velocity on the amplitude response of a square cylinder is well studied by introducing five different linear springs and consequently, the output behavior of SC-TENG as well. I therefore recommend this manuscript to be accepted for publication in Nanomaterials after minor revision of the following points.
1. I suggest the authors provide more explanation for the comparison of potential output performance between square grid structure-based and honeycomb structure-based SC-TENGs that was used in this study, in the identical geometry (i.e., 150 × 30 × 30 mm).
2. There could be an effect of the size of electronegative PTFE balls on voltage, current, and transferred charge. I recommend the authors provide more information about the relationship between PTFE ball sizes and the outputs.
3. I would suggest the authors discuss the Reynolds number in the wider range for more understanding on the VIV-based SC-TENG system.
Author Response
We would like to thank you first for all the positive comments of ourmanuscript (No.: 1946993) entitled “Experimental investigation of Reynolds Number and Spring Stiffness Effects on Vortex-Induced Vibration Driven Wind Energy Harvesting Triboelectric Nanogenerator”. We really appreciate your help and patience. We also think highly of the comments of both reviewers who kindly provide professional suggestions on our manuscript. We have seriously thought about them and provided our response to reviewers. For the detailed response, please see the attachment.

Reviewer 2 Report
In this work, a VIV driven square cylinder triboelectric nanogenerator (SC-TENG) was proposed for harvesting broadband wind energy. The vibration characteristic and output performance were studied experimentally to estimate the effect of natural frequency. The focus was on identifying the effect of natural frequency on the behavior and output performance of the SC-TENG undergoing VIV, by employing five different spring stiffness. This work deserves attention. My main complaints relate to the presentation of the results. The Introduction section is described logically and in detail (36 refs out of 40), however, the Materials and Methods section looks very poor, not enough description and details are provided. It needs to be improved as well as figures 3 and 5.
Author Response
Thank you for your kind comments on our manuscript entitled “Experimental investigation of Reynolds Number and Spring Stiffness Effects on Vortex-Induced Vibration Driven Wind Energy Harvesting Triboelectric Nanogenerator”. We have carefully revised the manuscript according to your valuable comments. Based on the suggestions, we have made an extensive modification on the Materials and Methods part, as well as Figures 3 and 5. The changes to our manuscript within the document were also highlighted by using red colored text. The detailes can be found in the revised manuscript.

Reviewer 3 Report
Experimental investigation of Reynolds Number and Spring Stiffness Effects on Vortex-Induced Vibration Driven Wind Energy Harvesting Triboelectric Nanogenerator is a nice work presented by Chang et al. They demonstrated a charging of capacitors. they also displayed some Comsol simulation results, which is interesting. They used it to power some thermometers from wind energy-driven TENG. After careful evaluation, I suggest a minor revision:
1. In the introduction, some of the important results and new work in triboelectric generators should be discussed: Materials Letters 304, 130674, 2021 and Nano Energy 101, 107620, 2022.
2. The authors should add a comparison table to show the performance enhancement to the proposed TENG with reported wind-driven TENG.
3. The long-term stability of the TENG device is missing?
4. Authors can show some more experimental demonstrations like the glowing of LEDs?
5. English grammatical errors need to be removed.
Author Response
Firstly, we would like to thank you for your kind letter and constructive comments concerning our article (Manuscript No.: 1946993). These comments are all valuable and helpful for improving our article. All the authors have seriously discussed about all these comments. According to your valuable comments, we have tried best to modify our manuscript to meet with the requirements of the journal. In this revised version, changes to our manuscript within the document were all highlighted by using red colored text. Point-by-point responses to the reviewers can be seen in the atttachment.
